# Estimation of Tool Life in the Milling Process—Testing Regression Models

**DOI:** 10.3390/s23239346

**Published:** 2023-11-23

**Authors:** Andrzej Paszkiewicz, Grzegorz Piecuch, Tomasz Żabiński, Marek Bolanowski, Mateusz Salach, Dariusz Rączka

**Affiliations:** 1Department of Complex Systems, Faculty of Electrical and Computer Engineering, Rzeszow University of Technology, al. Powstancow Warszawy 12, 35-959 Rzeszow, Poland; marekb@prz.edu.pl (M.B.); m.salach@prz.edu.pl (M.S.); 2Department of Computer and Control Engineering, Faculty of Electrical and Computer Engineering, Rzeszow University of Technology, al. Powstancow Warszawy 12, 35-959 Rzeszow, Poland; gpiecuch@prz.edu.pl (G.P.); tomz@prz.edu.pl (T.Ż.); 3Faculty of Electrical and Computer Engineering, Rzeszow University of Technology, al. Powstancow Warszawy 12, 35-959 Rzeszow, Poland; draczka@prz.edu.pl

**Keywords:** IIoT, CNC machine, machine learning, smart manufacturing, tool condition monitoring

## Abstract

The article presents an attempt to identify an appropriate regression model for the estimation of cutting tool lifespan in the milling process based on the analysis of the R^2^ parameters of these models. The work is based on our own experiments and the accumulated database (which we make available for further use). The study uses a Haas VF-1 milling machine equipped with vibration sensors and based on a Beckhoff PLC data collector. As the acquired sensor data are continuous, and in order to account for dependencies between them, regression models were used. Support Vector Regression (SVR), decision trees and neural networks were tested during the work. The results obtained show that the best prediction results with the lowest error values were obtained for two-dimensional neural networks using the LBFGS solver (93.9%). Very similar results were also obtained for SVR (93.4%). The research carried out is related to the realisation of intelligent manufacturing dedicated to Industry 4.0 in the field of monitoring production processes, planning service downtime and reducing the level of losses resulting from damage to materials, semi-finished products and tools.

## 1. Introduction

The development of modern industry, especially with regard to the Industry 4.0 concept, requires the interaction of manufacturing machines, sensors and specialised software to ensure greater efficiency, productivity and reliability of production processes. Ensuring the continuity of these processes is a significant challenge for industry [1,2,3]. The most typical example is bearing diagnostics. This subject has been continuously developed for decades [4,5,6,7], but there are fewer solutions in the context of real-time process monitoring. New ideas and solutions must be sought and implemented in order to achieve the desired goals in this area. One important aspect of ensuring the continuity of production processes is the prediction of wear and tear on the tools used. The comparisons included in [8,9,10,11,12] demonstrate how important and current this topic is. Proper estimation makes it possible to reduce costs related to possible unplanned production downtime as well as costs related to wasted material/semi-finished products due to damage to individual tools or machines [13].

The current and well-established realm of research pertains to the investigation of forecasting the extent of tool wear during the execution of technological operations, including drilling and milling. A literature analysis indicates that experiments are predominantly conducted using numerically controlled or conventional machine tools with various alloys of steel, aluminium, or Inconel as the processed materials. Numerous authors endeavour to devise methodologies for diagnosing cutting processes and predicting tool deterioration through the utilisation of machine learning [14,15].

Anticipating the wear process of a tool until it becomes unsuitable for use, known as Remaining Useful Life (RUL), permits the estimation of potential failure or damage time, thus minimising unforeseen downtime. This is particularly significant, considering that tool damage accounts for 7% to 20% of machine downtime and tool replacement contributes to 3% to 12% of production costs [16,17]. Employing an IT system built upon the Tool Condition Monitoring (TCM) approach proves instrumental in curtailing costs, reducing downtime, and optimising tool usage. Such activities are particularly important in the aviation and automotive industries, which are characterised by multi-series production focused on efficiency while maintaining a very high quality of production. This relates to reducing unplanned downtime and the waste of usually expensive materials and semi-finished products. Developing a solution based on the integration of manufacturing devices, sensors, and a data collection and processing system that will allow the continuous supervision of the manufacturing process is important. Moreover, the work carried out allowed for the preparation of a large standardised set of data that can be shared, which can be used to conduct further research work.

By analysing sensory data and the operating parameters of a device, artificial intelligence algorithms can contribute to the earlier detection of signs of future problems or failures [18,19,20,21]. This makes it possible to take preventive action, which in turn increases system reliability and saves costs. To classify tool conditions or predict failure instances, various signals from sensors, process variables, and images are harnessed. Accelerometers, force sensors, spindle current, work table drive axes, and acoustic emission sensors are the most commonly utilised. These measurement signals are assessed individually or combined, considering the intricacy of the problem. In particular, regression methods are often used for equipment damage prediction because of their ability to model relationships between different variables [22]. In the case of equipment damage prediction, we often deal with continuous data, such as equipment operating parameters, operating time or sensory values, and regression methods allow us to model these relationships and predict continuous values. Also of great importance is the fact that regression algorithms can be trained on historical data mapping specific operating conditions as well as patterns that are specific to the device. Of course, the models can be updated as a result of new data being provided. It is also not insignificant that regression models can be effectively trained and implemented in environments where a lot of data are received, providing scalability, including in continuous production environments. Therefore, by analysing the correlation between the signal and tool wear across time, frequency, and time–frequency domains, conventional metrics like RMS are determined and employed in shallow (ANN, SVM, HMM, SVR, LR) and deep (RNN, LSTM, CNN) classification and regression models.

In the manufacturing industry, there is a significant demand for intelligent production systems tasked with monitoring, for example, the wear progression of tools. These systems are expected to alert operators in real time based on the gathered data about potential failure or quality discrepancies in the product [23]. During the implementation of such systems, decisions are made regarding which signals to measure and how they should be collected and processed. The authors of [24] indicate that vibrations directly reflect the operational state of devices, considering the mechanical behaviour of devices, such as motor rotation, flow rates in pipes, etc. They consider vibration to be independent of other external factors such as temperature and humidity. They also propose the hypothesis that machine vibration effectively reflects the ageing process of devices, making it an excellent indicator of early failure symptoms. In [25], it is added that machine vibration analysis brings many benefits, including increased revenue, production reliability, shorter downtime, and lower repair costs.

Vibration levels generated by individual machine components are most commonly recorded, but signals such as acoustic emissions, power consumption, forces, temperatures, etc., are also often collected. With the increasing number of recorded signals, which can be several dozen on a single machine, there is a growing demand for efficient information systems capable of handling large amounts of streaming data (big data). Researchers and practitioners refer to such a model as 4V—volume, velocity, variety, and veracity [26,27]. A large dataset size also means that transferring data for processing is uneconomical; instead, calculations should be performed on the same machine where the data are stored. They also highlight the need to build analytical platforms for diagnostics based on additional sensors, given that factory-installed sensors typically provide data only for internal control systems. Moreover, in critical machine parts, pre-installed sensors are often lacking [26]. As the number of data increases exponentially with the development of IoT technology [28], research is being conducted to develop methods that maintain good classification quality while reducing the amount of data used in the learning process [29,30].

The methods of computational intelligence are utilised both for diagnosing and predicting the wear of CNC machine components, engines, gear assemblies, wind turbines, transformers, and many other devices. In practice, however, it is challenging to encounter a functioning real-world diagnostic or predictive system, which is largely due to the uncertainty in assessing and predicting the equipment’s condition and/or high initial investment costs. Much of the ongoing research in this field remains theoretical. Most often, these studies are conducted based on laboratory objects or publicly available datasets. There is a lack of research on the production reality and evidence of efficiency improvement based on the solutions described in the literature. Thus, the aim of this study is to verify the R^2^ parameter of regression models for estimating tool life in the milling process based on real data from the production environment. This article presents sample results of the cutting tool life estimation for a real CNC machine and a dataset collected in house, which we share more widely.

The article consists of several sections. Section 2 describes the laboratory bench built and used during the research, consisting of a three-axis Haas VF-1 milling machine and expanded to include additional vibration sensors, specifically accelerometers and current transducers. This stand enabled the collection of data necessary for the machine learning algorithms. Section 3 contains the characteristics of the regression models used, together with the parameters considered, as well as the results obtained from the tests carried out.

## 2. Testbed and Data Description

The research stand involves working with a three-axis Haas VF-1 milling machine. This specific machine is equipped with a proprietary measurement system and is augmented by additional vibration sensors, specifically accelerometers and current transducers (Figure 1). 

The VF-1 milling machine integrates a meticulously designed measurement system based on industrial automation components: Industrial PC (IPC) Beckhoff C6920, along with a combination of I/O modules, specifically EL1008, EL2008, EL3632 (4 units), EL3702, and EL3413 (4 units). The software driving this system is TwinCAT 3.1.

The measurement system includes strategically positioned sensors:A total of eight accelerometers (sensitivity 100 mV/g)—Dytran 3055A2 and one Dytran 3056D2T. Three of them on the spindle (Figure 1—Ch1: +Y axis, Ch2: −Z axis, Ch3: −X axis) and five sensors on the drives of the work table were mounted (for X direction Ch4: +Z axis, Ch5: −X axis and for Y direction Ch6: +Z axis, Ch7: +Y axis, Ch8: −X axis).A total of twelve current transducers—Wago 855-4001/0100-0001, accompanied by Wago 2007-8875 terminal blocks, are also integral to the system and were mounted in the electrical cabinet for each phase (L1, L2, L3) for the spindle and the X, Y, and Z axes.

When designing the experiments, consideration was given to other studies [31,32,33,34,35,36,37,38] to ensure the comparability of results. Experiments similar to previous studies but with a larger number of workpieces were proposed. The choice of material was influenced by its widespread use in the aviation industry [39] and the automotive industry [40,41,42] as well as the fact that steel workpieces, unlike aluminium, accelerate tool wear. Material 42 CrMo4 with manufacturing tolerance IT12 and hardness 38 ± 2 HRC (thermally hardened) was selected, although not all workpieces met the specified hardness parameter. Hardness measurements ranged from 35.33 to 41.67 HRC. Cuboid samples were utilised to facilitate clamping in the machine vice with dimensions of 80 mm × 80 mm × 150 mm chosen to ensure secure fastening to a depth of approximately 20 mm (Figure 2).

The key to conducting the experiments properly was continuous milling without retracting the tool from the material. To reflect real-life conditions and adapt to industrial machining, the milling path followed a clockwise movement around the workpiece contour. Due to the specificity of the milling process, the vibrations, power consumption, and roughness obtained differ depending on whether the milling is co-rotating or counter-rotating. In this case (omnidirectional movement), both types of milling occurred, so the direction of machining CW or CCW does not matter; only the order is different.

Some experiments used cutting parameters recommended by technologists, while others used unfavourable parameters. This was intended to accelerate the degradation of cutters as well as to examine the negative impact of changing milling parameters on their service life. The side recess of the cutter in the material was a crucial parameter with the manufacturer’s recommendation limiting it to 45% of the tool’s diameter. This allowed for 4 cycles per layer with 80% cutter insertion (non-optimal) and 7 cycles per layer with 45% cutter insertion (optimal) for a sample of this size (RDOC—Radial Depth of Cut values were 4.5 and 8 mm). The number of milling cycles also depended on the Axial Depth of Cut (ADOC), resulting in 48 to 84 cycles at ADOC = 5 mm and 24 to 42 cycles at ADOC = 10 mm. Two types of 10 mm diameter cutters were used: high quality (Van Hoorn VHVTR 4 100 070 10 03 050) and medium quality (PARA Tooling RS4 10.0x70). The parameter values for both cutters are: d = 10 mm (diameter), L = 70 mm (total length), L1 = 22 mm (cutting length), and z = 4 (number of flutes). Additionally, the impact of the tool holder length on system vibration was compared in the tests (two different lengths).

The collected data were labelled by sample number (P), layer number (F) and cycle number (C). Each sample was a single cuboid block of the material being processed, which was divided into layers corresponding to the thickness of material removed at each stage of the experiments. Cycles, each consisting of a single machining program run clockwise, were distinguished within the layers. The process of milling is illustrated in Figure 3. The number of layers and cycles for a given sample depended on the adopted ADOC and RDOC values. The basic process parameters were as follows: spindle speed: 3200 rpm, feed rate 640 mm/min, and tool holder length: 80 and 160 mm. Other process data and details are described on https://datasets.kia.prz.edu.pl/ (accessed on 17 October 2023).

Table 1 contains all statistical data regarding working time and the amount of material collected by individual cutters. Tool ID numbers from 0 to 100 apply to Van Hoorn cutters and over 100 apply to PARA Tooling cutters. Tools with ID 10 and 11 in the 160 mm tool holder length were mounted (for tool no. 11, the breakage was immediate). The collected data (Table 1) show that the RDOC parameter significantly affects the tool life, which directly affects the number of cycles performed. Based on the data, it was also observed that the ADOC parameter has no noticeable impact on the cutter lifespan. Therefore, assuming the same number of cycles possible for ADOCs of 5 and 10 mm, the volume of material collected from the sample will be twice as large with an ADOC of 10 mm. The conclusion is consistent with the HEM strategy (High-Efficiency Milling) [43], which consists of using the entire tool length (larger ADOC value) at the expense of reducing the radial depth of cut. The advantage of this milling strategy is the possibility of greater heat distribution in the tool, which increases its service life. For three PARA Tooling cutters, a decrease in the number of cycles performed was visible while maintaining optimal process parameters.

## 3. Preparation and Testing of Regression Models

For each of the tested learning algorithms, multiple models were created with different parameters specific to each algorithm. To achieve this, the GridSearchCV function from the sklearn library was employed as it is one of the default methods to optimise hyper-parameters in the sklearn library. To prevent overfitting during training, a 10-fold cross-validation was used. The training data subset was randomly divided into 10 equal parts, and the model with the specified parameters was trained 10 times, each time using a different subset as the validation set.

The evaluation metric used to assess the quality of the models was the R^2^ score, which takes a value of 1.0 in the case of a perfect result. The closer the quality metric value is to 0.0, the lower the accuracy of the created model. Finally, the best model was selected for each algorithm (the model with the best parameter set) and tested on a separate 20% test dataset.

### 3.1. SVR

The research was conducted using the SVR (Support Vector Regression) function from the sklearn library. It is a regression technique based on support vector machines (SVMs) that allows the modelling and prediction of continuous values using C and epsilon-free parameters. As a result of the initial analysis in machine learning, the utilisation of a kernel based on the radial basis function proved to be the most effective. This kernel allows for the transformation of data into a space of infinite dimensionality. The radial basis function (RBF) Is the default kernel method in the sklearn library’s Support Vector Regression (SVR). Since the initial learning results for other kernel functions were unsatisfactory, the analysis focused solely on the radial basis function.

To find the optimal model, 271,441 different combinations of the following parameters were examined [44]:Parameter C—This is a regularisation parameter that controls the importance of allowable errors. A higher value of this parameter gives more weight to errors (loose values falling outside the decision boundaries of the model), leading to more flexible and complex models (the model will try harder to fit the data). However, an excessively high value of this parameter can lead to overfitting. The default value of parameter C is 1.0. In the conducted experiments, 521 values were tested in the range from 10^−9^ to 10^4^, which were evenly spaced on a logarithmic scale.Parameter gamma—This is a kernel parameter that determines how far the influence of a single data point reaches on the model’s prediction. A high value of the gamma parameter makes points close to each other have a greater influence on the prediction, while a low value of this parameter emphasises points that are far apart. In the sklearn library, the default value of the gamma parameter is calculated according to the formula: 1/(number of features × variance). In the conducted experiments, 521 values were tested in the range from 10^−4^ to 10^9^, which were evenly spaced on a logarithmic scale.

Figure 4 presents the results of the obtained learning for the SVR for both the full range of tested parameters and the range of best results obtained.

It can be observed that the best result was achieved with a gamma parameter of approximately 13.3352 and a C parameter of approximately 501.187. For this parameter combination, the R^2^ score was 0.907 ± 0.119 (double standard deviation), which translates to approximately 90% prediction accuracy. When the C parameter was smaller than approximately 10^−3^ and the gamma parameter was larger than 10^5^, the learning results were consistently low. It is also noticeable that the gamma parameter has a greater influence on the learning results (for the C parameter, the learning results remained nearly constant after reaching a certain threshold). These observations and the learning results depending on individual parameters (for the best combination) are presented in Figure 5.

### 3.2. Regression Decision Tree

The research was conducted using the DecisionTreeRegressor function from the sklearn library. It is the default method for fitting regression models using decision trees, which are flexible tools for modelling nonlinear relationships and are widely used in regression tasks. To find the optimal model, 77,616 different combinations of the following attributes and parameters [45] were tested:Splitting criterion (criterion parameter)—describing the function used to measure the quality of a split. The following functions were tested:
Mean Squared Error (“squared_error”)—minimises the Mean Squared Error between predicted and actual values. It is calculated as the arithmetic mean of the squared prediction errors for each sample in the dataset.Friedman Mean Squared Error (“friedman_mse”)—operates similarly to Mean Squared Error but incorporates a Friedman correction to minimise the impact of noise on prediction results. Mean Absolute Error (“absolute_error”)—minimises the Mean Absolute Error between predicted and actual values.Poisson Deviance (“poisson”)—uses Poisson deviance reduction to find appropriate splits.Splitting strategy at each node (“splitter”)—two options were available within the algorithm, both of which were tested:
Best (“best”)—selects the best possible split.Random (“random”)—selects the best random split. Minimum number of samples required to split a node (“min_samples_split”)—values were tested in the range from 2 to 99.Minimum number of samples required to create a leaf node (“min_samples_leaf”)—values were tested in the range from 1 to 99.

In the case of decision tree utilisation, the best learning results were achieved using the criterion based on Poisson deviance. The trained model exhibited approximately 92% accuracy with an R^2^ parameter of 0.915 ± 0.103 (double standard deviation). Similar learning results were obtained with criteria based on the Mean Squared Error; almost identical values were achieved in both cases. The criterion based on mean absolute error performed significantly worse in learning. 

The choice of different splitting strategies did not affect the best result achieved by individual criteria. However, it is noticeable that the best split strategy resulted in better prediction results across the entire range of tested parameters. The experiments showed that in each of the tested cases, the best learning results were achieved with a relatively small number of samples required to split a node or create a decision tree leaf node.

### 3.3. Regression by One-, Two- and Three-Layer Neural Networks

The research was conducted using the MLPRegressor function from the sklearn library. It is the default method for fitting a regression model using a multi-layer perceptron, which allows the modelling of complex nonlinear relationships between features and the explanatory variable. To find the optimal artificial neural network model, a total of 195,088 different combinations of learning parameters were tested. The parameters tested included the following [46]:Activation function parameter—defining the activation function used in hidden layers. All available activation functions in the sklearn library were tested (identity, logistic, tanh, ReLU).Solver parameter—determining the optimisation algorithm used to minimise the cost function and find the appropriate set of weights in the neural network. The Adam and LBFGS solvers were tested.Number of neurons in hidden layers:
For single-layer networks—the number of neurons in the hidden layer was tested from 1 to 496 with a step of 5 (a total of 100 possibilities).For two-layer networks—the number of neurons in individual layers was tested from 2 to 299 with a step of 3 (a total of 10,000 combinations).For three-layer networks—the number of neurons in individual layers was tested from 5 to 205 with a step of 10 (a total of 9261 combinations).For single-layer networks, the influence of the alpha parameter on network learning was also tested. This parameter determines the regularisation penalty added to the cost function and directly affects the regularisation effect. The default value of the alpha parameter in the sklearn library is 10^−4^. In the conducted experiments, 51 values were tested in the range from 10^−5^ to 10^5^, which were evenly spaced on a logarithmic scale.

#### 3.3.1. Single-Layer Network

In the case of single-layer networks, the best learning results were achieved using the ReLU activation function and the LBFGS solver. The best-trained model exhibited approximately 96% accuracy with an R^2^ parameter of 0.962 ± 0.085 (double standard deviation). When using the Adam solver, the best result was also achieved with the ReLU activation function, but the learning accuracy was much lower, around 73%, with an R^2^ parameter of 0.728 ± 0.214.

Satisfactory learning results were also achieved with the LBFGS solver when using the hyperbolic tangent function (for fewer than 100 neurons) and the logistic function (for fewer than 20 neurons). In other cases, the prediction accuracy did not exceed 70%. For all learning combinations, it was observed that the alpha parameter had a nearly uniform impact. Results rapidly decreased to almost zero for values from 1 to 10 and remained at that level until the maximum tested value. The best results were observed for small values of the alpha parameter (values less than 1). The aggregate learning results for different activation functions and the LBFGS solver are presented in Figure 6.

#### 3.3.2. Two-Layer Network

In the case of two-layer networks, similar to single-layer networks, the best learning results were achieved using the ReLU and Tanh activation functions with the LBFGS solver. The best models exhibited approximately 96% accuracy with an R^2^ parameter of 0.959 ± 0.065 for ReLU and 0.959 ± 0.034 for Tanh (Figure 7). When using the Adam solver, the best result was also achieved with the ReLU activation function, but the learning accuracy was lower, around 88%, with an R^2^ parameter of 0.877 ± 0.104.

#### 3.3.3. Three-Layer Network

For both solvers, the identity activation function resulted in only approximately 65% learning accuracy across the entire range of tested parameters. Using the LBFGS solver, the results obtained by using a linear activation function differed only within 0.001 (Figure 8a,b), which corresponds to the results obtained for one-layer and two-layer networks. From these results, we can conclude that there are no linear relationships between the input data. 

In the tested cases, the logistic activation function performed poorly with exceptionally low learning results (Figure 8c). The vast majority of results did not exceed an accuracy of more than 1%. There were a few combinations of parameters when using the LBFGS solver that achieved higher learning results (up to around 90%—Figure 8d). Higher regression results were achieved for neural networks with fewer neurons in each layer (as in the cases of single-layer and two-layer networks); however, this may have been achieved by randomly fitting the test data to the model (bias). The results obtained confirm the information found in the sources that the logistic function is definitely better suited in the context of classification problems than for regression problems.

Using the Adam solver, the hyperbolic tangent function returned results similar to those obtained with the identity function. The LBFGS solver performed significantly better, allowing for a maximum learning accuracy of approximately 96% with an R^2^ parameter of 0.96 ± 0.052 being the best-achieved value for the tested three-layer networks. The ReLU activation function consistently allowed for high learning results. For the LBFGS solver, the maximum accuracy achieved with this function was approximately 96% with an R^2^ parameter of 0.958 ± 0.052. For the Adam solver, the highest accuracy was around 93% with an R^2^ parameter of 0.93 ± 0.073. The aggregate learning results for different activation functions and the LBFGS solver are presented in Figure 8.

## 4. Discussion

The problem lies in selecting an appropriate classification or regression model for the dataset under consideration. If correct assumptions about the dataset are not made, no computational intelligence algorithm can be deemed superior to others as a consequence of the No Free Lunch Theorem. It often happens that data or their features exhibit a linear nature, making the application of linear models most appropriate. On the other hand, for more complex data, the use of models built on neural networks, for example, might be more suitable. Therefore, for each dataset, several to several dozen algorithms should be appropriately selected and tested. 

The best results obtained on the learning dataset were achieved for one-, two- and three-layer neural networks using the LBFGS solver. These results were characterised by the lowest error values. In the case of the Adam solver, it was observed that increasing the number of layers positively impacted learning accuracy. For support vector machines and decision trees, similar results were achieved with accuracy levels exceeding 90%. A summary of the results achieved for each of the machine learning methods is shown in Figure 9.

The machine learning models obtained were then tested on a 20% randomly selected test dataset, which was separated before the learning process. This step was aimed at checking that the models were not overfitting to the training data. The obtained results are presented in Figure 10.

The obtained prediction results have similar levels to the predictions achieved on the training datasets and fall within the range of the error obtained through 10-fold cross-validation. Considering that the created models have not had prior access to the input data on which the final validation was carried out, we can exclude the occurrence of overfitting.

With the exception of single-layer neural networks using the Adam solver, all machine learning algorithms enabled the achievement of satisfactory and high-quality prediction models.

The fastest learning algorithms were decision trees and the support vector machine algorithm. In the case of artificial neural networks, the learning speed was inversely proportional to the number of network layers.

A comparative analysis of the various machine learning regression models made it possible to show how they behave in a specific context, in this case, in predicting the failure of a milling machine. This allows the selection of the best method or approach to a given problem, which can significantly contribute to improving the efficiency and quality of industrial processes. 

The models created would achieve very high learning efficiency. Referring to the literature and other studies that assume the prediction of the life cycle of a tool, it can be considered that the results achieving an accuracy level of more than 90% are very satisfactory. However, comparisons of this type are not conclusive, since it is impossible to objectively compare models based on different learning data; therefore, the effectiveness of each model should be evaluated individually. Sometimes, due to the high complexity of the data, it may not be possible to achieve such high accuracy as in the case we studied, but this does not mean that such results are not equally valuable.

The use of artificial intelligence in industry is of great importance in the context of improving efficiency, optimising costs and improving the safety of production processes. The proper selection and optimisation of machine learning methods is one of the key aspects of creating effective solutions.

The conducted research allows us to understand what factors affect the effectiveness of individual methods. By performing a series of tests, it was possible to optimise learning parameters, which can be a valuable indication for engineers and specialists in the field of industrial process automation. Improperly selected learning parameters can prolong the learning process even several times without providing any benefit in the achieved results.

In addition, by using machine learning models to monitor and predict failures, companies can introduce a proactive approach to equipment maintenance. This makes it possible to avoid prolonged production downtime and costly repairs, which significantly contributes to saving time and resources.

This research is also crucial for the continuous improvement of industrial processes. Based on the results of benchmarking, machine learning models can be adjusted and improved, leading to more accurate predictions and the more efficient use of production resources.

It is worth noting that the artificial intelligence-based solutions developed can be adapted and scaled to different types of industrial machinery and equipment, making them versatile tools in the process of automation and optimisation in industry.

## 5. Conclusions

The use of artificial intelligence in industry is becoming increasingly widespread. As the performance of the prediction mechanisms is strictly dependent on the adopted training dataset, a significant aspect of the conducted work involved constructing an appropriate dataset under industrial conditions. These data formed the basis for further research regarding the application of regression methods for estimating tool life in the milling process. As a result of the conducted experiments, the following conclusions were drawn:The best results for both the training and testing datasets were achieved using two-layer neural networks with the LBFGS solver (95.9% and 93.9%, respectively).The worst results were obtained for both datasets in the case of one-layer neural networks using the Adam solver (72.8% for the training dataset and 63.4% for the testing dataset).Other employed models yielded results around 90% for both datasets.

Taking into account the obtained results, it can be assumed that regression models allow the estimation of cutting tool wear in the milling process. The information obtained in this way can be used in the IT system to plan production downtime, but above all, it can lead to counteracting uncontrolled failures and damage to manufactured components and semi-finished products. Such activities are particularly important in the aviation and automotive industries, where materials and manufactured elements are often expensive and, moreover, must meet high-quality standards. Therefore, elements manufactured with possible defects resulting from faulty tools are not allowed. The use of tested models, especially neural networks, may contribute to the elimination or significant reduction in these undesirable situations.

## Figures and Tables

**Figure 1 sensors-23-09346-f001:**
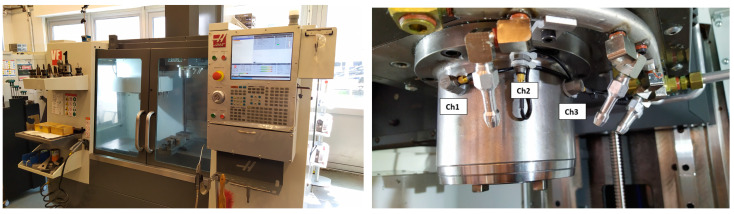
Haas VF-1 milling machine and mounting location of 3 sensors on the spindle.

**Figure 2 sensors-23-09346-f002:**
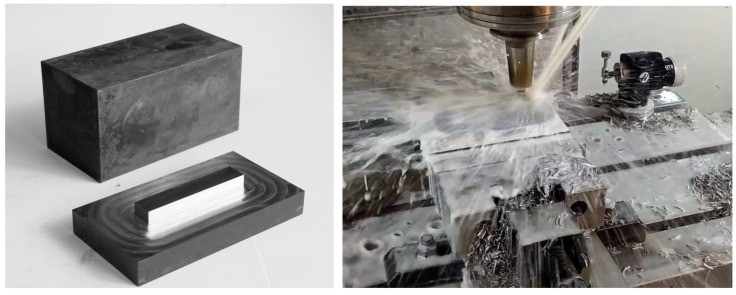
Samples before and after milling (on the **left**) and during the milling (on the **right**).

**Figure 3 sensors-23-09346-f003:**
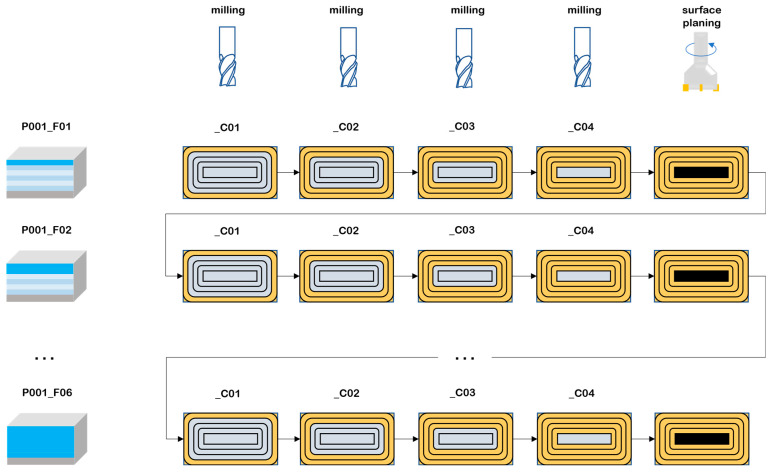
Process of conducting experiments.

**Figure 4 sensors-23-09346-f004:**
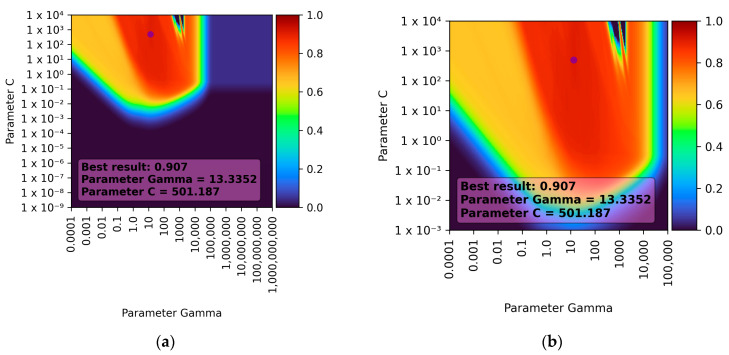
Visualisation of teaching results for the SVR model: (**a**) full range of tested parameters; (**b**) range of best results.

**Figure 5 sensors-23-09346-f005:**
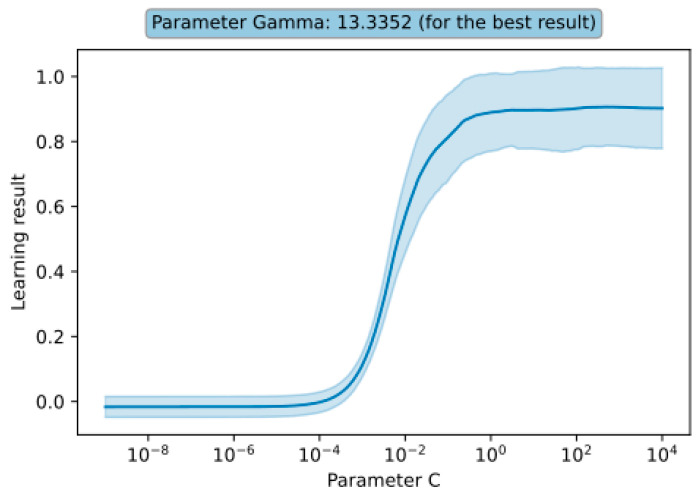
Results of teaching the SVR model depending on the C parameter.

**Figure 6 sensors-23-09346-f006:**
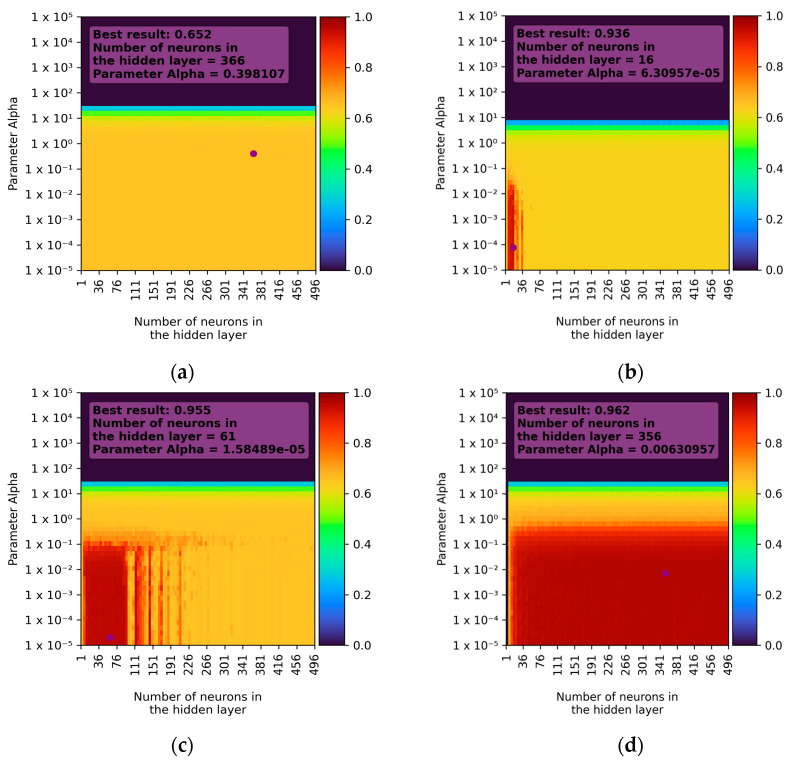
Example of learning results for the LBFGS solver depending on activation function (single-layer network): (**a**) activation function—identity; (**b**) activation function—logistic; (**c**) activation function—Tanh; (**d**) activation function—ReLU.

**Figure 7 sensors-23-09346-f007:**
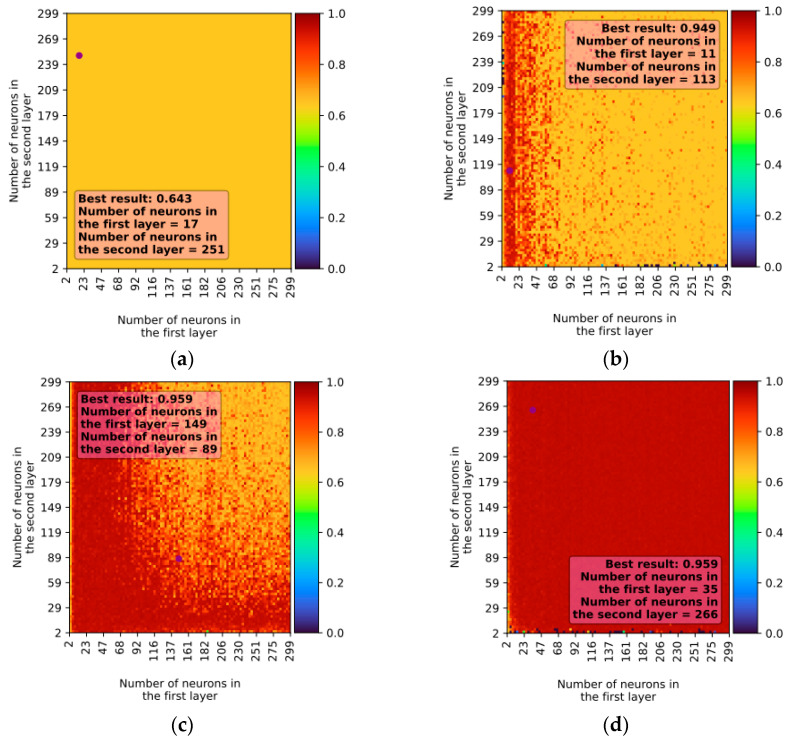
Example of learning results for the LBFGS solver depending on activation function (two-layer network): (**a**) activation function—identity; (**b**) activation function—logistic; (**c**) Activation function—Tanh; (**d**) activation function—ReLU.

**Figure 8 sensors-23-09346-f008:**
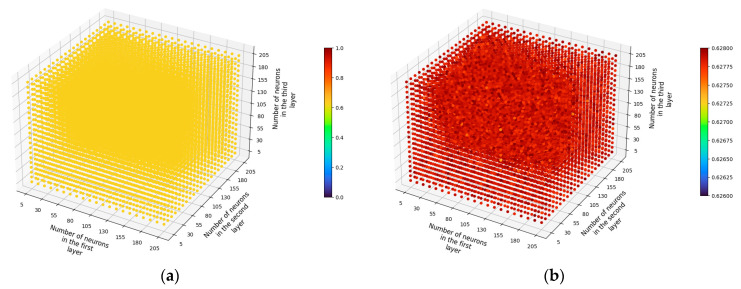
Example of learning results for the LBFGS solver depending on activation function (three-layer network): (**a**) activation function—identity; (**b**) activation function—identity, scale adjusted to results; (**c**) activation function—logistic; (**d**) activation function—logistic, accuracy greater than 1%; (**e**) activation function—Tanh; (**f**) activation function—ReLU.

**Figure 9 sensors-23-09346-f009:**
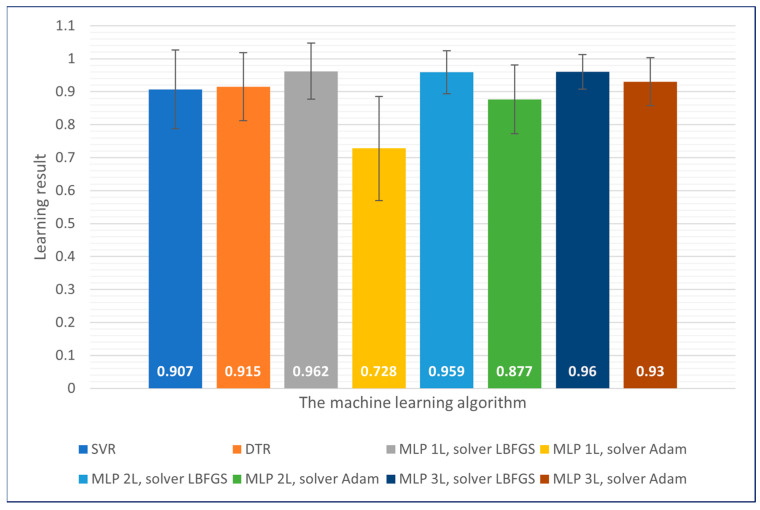
The results achieved for each of the machine learning methods—training dataset.

**Figure 10 sensors-23-09346-f010:**
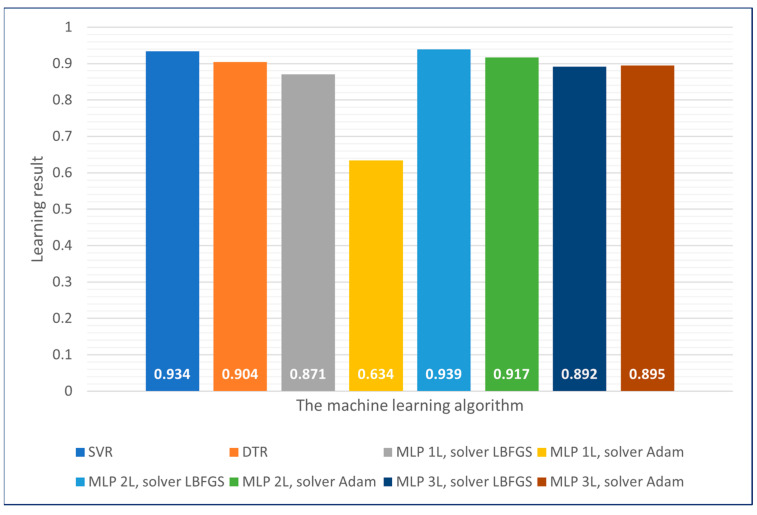
Results of the best models of individual machine learning algorithms—test dataset.

**Table 1 sensors-23-09346-t001:** Statistical data about tool lifetime—approximate values.

Tool ID	ADOC (mm)	RDOC (mm)	Number of Cycles Performed by the Tool	Total Tool Operating Time (s)	Total Length of the Tool Path (mm)	Total Volume of Milled Material (mm^3^)
2	5	8	49	1526	17,932	717,280
3	10	8	49	1526	17,932	1,434,560
4	5	8	40	1240	14,560	582,400
5	10	8	53	1605	19,388	1,551,040
6	5	8	29	906	10,652	426,080
7	10	4.5	126	3852	44,352	1,995,840
8	5	4.5	115	3531	40,696	915,660
9	5	4.5	149	4569	52,628	1,184,130
10	10	4.5	14	528	4928	221,760
11	10	4.5	0	0	0	0
101	10	4.5	83	2528	29,108	1,309,860
102	5	4.5	84	2568	29,568	665,280
103	10	4.5	59	1809	20,876	939,420
105	5	4.5	124	3812	43,892	987,570

## Data Availability

The data and photos are available publicly and free of charge on the website: https://datasets.kia.prz.edu.pl/.

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
