# Peer review of "Estimation of Tool Life in the Milling Process—Testing Regression Models"

_sensors, 2023, doi:10.3390/s23239346_

Round 1
Reviewer 1 Report
Comments and Suggestions for Authors
This manuscript attempts to propose a regression model for the estimation of cutting tool lifespan in the milling process, and the following suggestions are listed for references:
1. Cutting parameters should be included in the context. Currently cutting parameters such as spindle speed and feedrate are not included in the manuscript but in the linked web "www.datasets.kia.prz.edu.pl".
2. Other related parameters are suggested to be included as well, for example, number of flutes, tool helical angle, tool extension length, etc.
3. Indication on where the sensors such as the "accelerometers" and "current transducer" are installed should be added in Figure 1.
4. Verification of the estimated regression should be added. For example, if the cutting direction of toolpath is Counter Clockwise, can the proposed regression model be usable.
5. More description should be added for the explanation on Figure 8 (a), (b), (c), and (d). Currently, Figure 8 (a), (b), (c) are not clear pictures.
6. Explanation on why the AI algorithms are utilized could be explained for explainable AI application. For example, why the functions "GridSearchCV()", "DecisionTreeRegressor()", and "MLPRegressor() function" were selected?
Comments on the Quality of English LanguagePunctions and grammar could be enhanced to ease the readability of the manuscript.
Author Response
Responses to the reviewer's comments are provided in the attached file - Response_to_reviewer_1.pdf

Reviewer 2 Report
Comments and Suggestions for Authors
The authors presented an article about “Estimation of tool life in milling process – regression models test.”
In this study, the authors developed a regression model to predict tool life in the milling process. They have presented to readers a subject that has a significant place in the manufacturing industry. However, the article's presentation is at such a low level that it leaves behind the importance of the subject. The article is presented as a conference paper. I think the paper is not well organized and appropriate for the “Sensors” journal, but the paper will be ready for publication after major revision.
· The abstract do not looks good. Please include all significant numerical results. Also, please briefly describe the problem.
· What is the problem? Why was the manuscript written? Please explain the reason in the introduction part. In the last paragraph of the introduction, the novelty of the study and the differences from the past in detail should be expressed.
· Please the authors developed a regression model to predict tool life in the milling process. They have presented to readers a subject that has a significant place in the manufacturing industry. However, the article's presentation is at such a low level that it leaves behind the importance of the subject. The article is presented as a conference paper.
· Is Figure 2 a drawing made by the authors? Otherwise please cite.
· What do the authors mean in this sentence: "Some experiments used cutting parameters recommended by technologists, while others used unfavorable parameters." Please explain in more detail in the article.
· Two types of 10 mm diameter cutters were used: high-quality and medium-quality." The cutting tool description is not professional in this way. Please give the cutting tool specifications as a table.
· Line 133 R2 à R2 Please corret it. Line 205, etc.
· Line 153 “ … from 10^-9 to 10^4…” Line 160-172-234-235 etc. please correct. (10-9 and 104)
· The tool life of cutting tools is not mentioned. Not providing it in the article may create question marks in the reader's mind.
· It is generally recommended that the Discussion section be written comparatively based on literature sources. In this article, no comparison is made with reference to the literature at the end. This is a significant shortcoming. Please strengthen your results with literature information.
· In the Conclusion section, the essential numerical results of the article are also given. It is also recommended that you write in bullet points.
· Please fix the typographical and eventual language problems in the paper.
*** Authors must consider them properly before submitting the revised manuscript. A point-by-point reply is required when the revised files are submitted.
Comments on the Quality of English LanguagePlease fix the typographical and eventual language problems in the paper. (Extensive)
Author Response
Responses to the reviewer's comments are provided in the attached file - Response_to_reviewer_2.pdf

Round 2
Reviewer 1 Report
Comments and Suggestions for Authors
Thanks for responding to the comments and suggestion.
Comments on the Quality of English LanguageThe writings of the English manuscript is smooth for understanding.
Reviewer 2 Report
Comments and Suggestions for Authors
Thank you for revised manuscript
Comments on the Quality of English LanguageMinor editing of English language required